# Polyaniline-Derived N-Doped Ordered Mesoporous Carbon Thin Films: Efficient Catalysts towards Oxygen Reduction Reaction

**DOI:** 10.3390/polym12102382

**Published:** 2020-10-16

**Authors:** Javier Quílez-Bermejo, Emilia Morallón, Diego Cazorla-Amorós

**Affiliations:** 1Departamento de Química Inorgánica and Instituto de Materiales, Universidad de Alicante, Ap. 99, 03080 Alicante, Spain; javiq@ua.es; 2Departamento de Química Física and Instituto de Materiales, Universidad de Alicante, Ap. 99, 03080 Alicante, Spain; morallon@ua.es

**Keywords:** oxygen reduction reaction, polyaniline, N-doped carbon materials, ordered mesoporous carbon materials

## Abstract

One of the most challenging targets in oxygen reduction reaction (ORR) electrocatalysts based on N-doped carbon materials is the control of the pore structure and obtaining nanostructured thin films that can easily be incorporated on the current collector. The carbonization of nitrogen-containing polymers and the heat treatment of a mixture of carbon materials and nitrogen precursor are the most common methods for obtaining N-doped carbon materials. However, in this synthetic protocols, the surface area and pore distribution are not controlled. This work enables the preparation of 2D-ordered N-doped carbon materials through the carbonization of 2D polyaniline. For that purpose, aniline has been electropolymerized within the porous structure of two different templates (ordered mesoporous Silica and ordered mesoporous Titania thin films). Thus, aniline has been impregnated into the porous structure and subsequently electropolymerized by means of chronoamperometry at constant potential. The resultant samples were heat-treated at 900 °C with the aim of obtaining 2D N-doped carbon materials within the template structures. Polyaniline and polyaniline-derived carbon materials have been analyzed via XPS and TEM and characterized by electrochemical measurements. It is worth noting that the obtained 2D-ordered mesoporous N-doped carbon materials have proved to be highly active electrocatalysts for the ORR because of the formation of quaternary nitrogen species during the heat treatment.

## 1. Introduction

The oxygen reduction reaction (ORR), occurring in the cathode electrode of the fuel cells (FCs), is one of the main limitations of these devices because of the slow reaction rate and high overpotential of the reaction [1,2]. The development of new materials capable of increasing the kinetics of the ORR and being cheaper in comparison to the current commercial catalyst, based on platinum, is necessary for the industrial commercialization of these electrochemical devices [3,4]. In this regard, metal-free catalysts based on heteroatom-doped carbon materials arises as one of the most promising alternatives due to its highly efficient catalysis, low-cost, and long durability towards the ORR [5,6,7,8].

Among the heteroatom-doped carbon materials, it should be highlighted the use of nitrogen [9,10,11], phosphorus [12], sulphur [13,14], and oxygen [15,16,17], as doping agents. Among all these heteroatoms, nitrogen functionalization has been by far the most studied. Dai et al. [18] demonstrated the highly efficient catalysis of N-doped carbon nanotubes with platinum-like performance in alkaline medium. Nevertheless, there are three factors that limit the catalysis of the oxygen reduction reaction with N-doped carbon electrocatalysts [19]: (i) the chemical nature of the active sites, (ii) electrical conductivity, and (iii) the specific surface area and porous structure. The debate about the chemical nature of the active sites is now decreasing due to the synergy between experimental [20,21,22] and computational results [20,23,24] of some works, which point out the edge-type quaternary nitrogen and pyridine species in armchair position as the most active sites among nitrogen functionalities [20]. Moreover, carbon materials by themselves often exhibit high electrical conductivity. 

However, surface area and porous structure are also important properties, because they determine the accessibility of the active sites and the transport properties of the ORR intermediates through the pores to the active sites. Unfortunately, the most common preparation methods of N-doped carbon materials are based on the heat treatment of a mixture of nitrogen and carbon-containing precursors [25] or via direct carbonization of a nitrogen precursor [26], such as conducting polymers. The main problem of these routes of synthesis is the non-control over the porosity and surface area. 

In this sense, we report a method to synthesize N-doped ordered mesoporous carbon materials thin films with high control over the N active sites towards oxygen reduction reaction and maintaining a well-ordered porosity. To do this, a thin film of polyaniline is electropolymerized in Silica- and Titania-ordered mesoporous templates by aniline adsorption and subsequent polymerization at constant potential. Once the electropolymerization was performed, the samples were heat-treated with the aim of creating highly active species in the resultant composite nitrogen-doped mesoporous carbon material/template. This methodology permits an excellent control over the amount of N-doped carbon material and the chemical nature of the N species. The electrochemical behavior and physicochemical properties of the prepared samples were determined by XPS, TEM, and cyclic voltammetry in presence and absence of dioxygen in the alkaline electrolyte.

## 2. Experimental

### 2.1. Materials and Reagents

Aniline was purchased from Sigma Aldrich (St. Louis, MO, USA) and was distilled under reduced pressure prior its use, in order to remove the impurities (e.g., aniline oligomers formed by oxidation during the storage). Potassium hydroxide (KOH) was purchased from Sigma-Aldrich. Sulphuric acid (98% H_2_SO_4_) were purchased from VWR Chemicals (Radnor, PA, USA). Titanium tetraisopropoxide (TTIP), tetraethyl orthosilicate (TEOS), Pluronic123 (P123), and Pluronic^®^F127 were purchased in Sigma-Aldrich. All the solutions were prepared using ultrapure water (18 M cm from an Elga Labwater Purelab system, Wycombe, UK). The N_2_ (99.999%) and O_2_ (99.995%) was provided by Air Liquide (Paris, France) and were used without any further purification or treatment. The carbon support (G), consisting of commercial macroporous carbon sheets (thickness = 0.3 mm, mean pore size 0.7 μm) was provided by Poco Graphite (Decatur, TX, USA).

### 2.2. Silica Template

Nanostructured Silica thin films were successfully synthesized onto the graphite current collector and used as the hard template. Pluronic^®^F127 (2.52 g) was dissolved in absolute ethanol (40 mL) by stirring during 24 h under a controlled relative humidity (50%). TEOS (6.72 mL) was added and stirred during 1 h to a HCl solution (16.7 µL of HCl (37%) in 2.52 mL of water). Then, both solutions were mixed and stirred during 1 h under the same relative humidity. The final molar ratio of the mixed solution was 1 TEOS: 6.6·10^−3^ Pluronic©F127: 6.66·10^−3^ HCl: 4.62 H_2_O: 22.6 EtOH (that has been previously used to synthesize a rhombohedral silica thin film [27,28]).

The commercial graphite current collector sheet of around 1 × 1 cm^2^ was dip-coated in the mixed solution. The constant withdrawal rate was 60 mm·min^−1^ and the relative humidity was maintained at 50%. The film was aged at room temperature during 24 h. Then, the synthesized thin film was heat-treated at 450 °C for 5 h in a nitrogen atmosphere in order to eliminate the surfactant. The flow rate was maintained at 100 mL·min^−1^ and the heating rate was 1 °C·min^−1^. The final sample is named as G/Si.

### 2.3. Titania Template

Mesoporous TiO_2_ thin films were grown by a route found in the literature, which gives rise to hexagonal thin films [29,30]. In the synthetic protocol, 1.05 g of Pluronic 123 was added to 16.21 g of absolute ethanol. The solution was stirred until completely clear, and then 4.03 g of concentrated (37%) HCl was added dropwise. To the resulting mixture, 5.70 g of TTIP was added dropwise, giving the final solution the following molar composition: 1 TTIP: 0.01 P123: 17.6 EtOH: 1.9 HCl: 7.2 H_2_O (that has been previously used to synthesize a hexagonal Titania thin film [29,30]). Titania films were produced by dip-coating at a withdrawal speed of 60 mm·min^−1^ on the commercial graphite support (1 cm × 1 cm) and using the dip-coating procedure. The relative humidity (RH) during dip-coating was set at 50%. After deposition, films were aged at RH 50% for 24 h. Calcination of the films was performed in a furnace at 550 °C for 1 h under inert atmosphere (N_2_). In all experiments, the heating rate was 1 °C·min^−1^. The samples were named as G/Ti.

### 2.4. PANI Electropolymerization

The electrochemical polymerization was performed in a Biologic VSP potentiostat using a standard three-electrode cell configuration, with graphite as a counter electrode and Ag/AgCl/Cl^−^(sat.) electrode as a reference electrode. However, all potentials will be referred to the reversible hydrogen electrode (RHE).

Bare G, G/Si, and G/Ti were initially immersed into a solution of 0.1 M aniline + 0.5 M H_2_SO_4_ during 3 min to produce the adsorption of aniline monomer within the Silica and Titania pores. Then, the electrode was transferred to the electrochemical polymerization cell containing 0.5 M H_2_SO_4_ solution (without aniline) at a controlled potential of 0.2 V vs. RHE for 10 s. After this period, chronoamperometric polymerization was carried out at 0.85, 0.95, and 1.05 V vs. RHE for 5 min. While the polymerization does not take place at 0.85 V, the potential of 1.05 V is too high to reach control over the polyaniline electropolymerized. Therefore, the selected potential was 0.95 V vs. RHE and different polymerization times were used (10 s, 30 s and 5 min) with the aim of achieving different quantities of polyaniline. The resultant samples were referred as G/Si-PANI-X or G/Ti-PANI-X, where X is the time employed during the aniline polymerization. 

### 2.5. Heat Treatments

The samples were heat-treated in a tubular furnace at 900 °C for 1 h using a heating rate of 5 °C min^−1^ in an inert atmosphere (N_2_). The furnace was purged for 1 h before the heat treatment in the corresponding atmosphere; the flow rate was maintained at 100 mL·min^−1^ during the treatment. The samples were named as G/Si-PANI-X-900 and G/Ti-PANI-X-900, X being the time employed during the polymerization.

### 2.6. Physicochemical Characterization

The surface composition and oxidation states of the elements in the carbon materials were studied by XPS in a VG-Microtech Mutilab 3000 spectrometer and Al Kα radiation (1253.6 eV). The deconvolution of the XPS N1s region was done by least squares fitting using Gaussian–Lorentzian curves, while a Shirley line was used for the background determination. The samples were characterized by transmission electron microscopy (TEM) coupled to EDX with a JEOL JEM-2010 microscope operating at 200kV with a spatial resolution of 0.24 nm. FESEM images were obtained from ZEISS, Merlin VP Compact model coupled to EDX. XRD patterns were recorded for G/Ti samples, with Cu Kα radiation (0.1540 nm), at a scanning rate of 2°·min^−1^, in the 2θ range 5–80° (Miniflex II Rigaku (30kV/15 mA).

### 2.7. Electrochemical Characterization

Electrochemical characterization of all synthesized materials was performed in a 0.1 M H_2_SO_4_ and/or 0.1 M KOH solutions, using the three-electrode cell configuration. The electrochemical behavior was studied by cyclic voltammetry (CV) in a nitrogen-saturated solution between 0.0 and 0.8 V vs. RHE at 50 mV·s^−1^. A size of 1 × 1 cm^2^ of the electrode was immersed into the electrolyte as the working electrode. Graphite was used as a counter electrode and Ag/AgCl (3 M KCl) as the reference electrode. All the potentials are referred to the reversible hydrogen electrode.

The electrocatalytic activity towards oxygen reduction reaction was studied by linear sweep voltammetry (LSV) in O_2_-saturated 0.1 M KOH solutions between 1.0 and 0.0 V (vs. RHE). 

## 3. Results and Discussion

### 3.1. Electrochemical Polymerization and Characterization of G/Si-PANI and G/Ti-PANI Composites

The chronoamperograms recorded during the growth of polyaniline on G, G/Si, and G/Ti electrodes are presented in Figure 1a. The dashed red line shows the current for the bare macroporous graphite, the dotted blue line corresponds to the polymerization on G/Si, and the solid green line shows the growth of polyaniline on G/Ti surface. The current versus time curves observed are common responses for electrochemical deposition of polyaniline. A maximum of the current is observed for the three electrodes as a consequence of the consumption of aniline during the electrochemical polymerization, and then, the current decreases, as it corresponds to a diffusion process. The maximum current for G/Si is observed at approximately 5 s after the pulse, whereas the maximum for the polymerization over the pristine G and G/Ti needs more time (7 and 10 s, respectively). This fact reflects the different kinetics towards the aniline polymerization, being G/Si the most electroactive electrode. After 5 min of the potential pulse, the adsorbed aniline monomers are completely polymerized regardless of the electrode used. The measured charge together with the amount of deposited polyaniline after the potentiostatic polymerizations, determined as indicated in references [31,32], are shown in Table 1.

G/Ti material contains the higher amount of polyaniline. This fact should be related to the pore volume of the Titania thin film. The higher the pore volume, the higher the adsorption of aniline monomer and, consequently, the amount of electrodeposited polyaniline. As G is a macroporous material, the pore volume is low and, in comparison with those materials where the mesoporosity predominates (such as G/Ti and G/Si), the amount of obtained polyaniline is much lower. 

Figure 1b shows the steady voltammograms in 0.5 M H_2_SO_4_ solution of the electrodes prepared in Figure 1a after 5 min of polymerization. G/Si and G/Ti samples show the typical profile of polyaniline in sulphuric acid. Moreover, the voltammetric charge of these electrodes is higher than the equivalent electrode over G. This is due to the differences in surface area. Polyaniline grown in the Silica or Titania pores presents a higher electroactive surface than the electrode without the inorganic matrix. The peak corresponding to the leucoemeraldine–emeraldine transition appears at around 0.4 V vs. RHE [31,33], and the second peak can be associated to the redox process of quinone-type species as a consequence of the overoxidation of polyaniline [31,32].

In order to control the length of the polymer chain maintaining the mesoporous structure, G/Si and G/Ti electrodes were also subjected to 10 and 30 s of the potentiostatic pulse. After all polymerizations, the resulting G/Si-PANI and G/Ti-PANI samples were electrochemically characterized by cyclic voltammetry in 0.5 M H_2_SO_4_ solution (Figure 1c,d). For G/Si-PANI and G/Ti-PANI electropolymerized during 10s, a pair of reversible peaks at around 0.6 V is observed. The further growth of the aniline loading results in a shift of this reversible peak to higher potentials. This peak at 0.6 V observed for the lowest polyaniline loading can be associated to the response of a major product of aniline dimerization in acid media [34,35,36]. Thus, the shape of the voltammograms is governed by polymer content and becomes similar to the polyaniline for the samples with higher polyaniline content. This behavior has been previously observed for carbon/polyaniline composites [35].

Then, the time employed during the chronoamperometric experiment is directly related with the electrochemical response of the final material, which means that it is possible to have some control over the thickness of the deposited polyaniline and over the polymer length. While G/Si-PANI-30s and G/Si-PANI-5min clearly show the typical redox processes, G/Si-PANI-10s exhibits an important broadening of both peaks. This confirms the presence of small oligomers because of the short period of time employed during the polymerization, which does not enable the formation of large chains of polyaniline [35]. 

### 3.2. Electrochemical Characterization of Ordered Mesoporous N-Doped Carbon-Based Composites

All previously characterized G/Si-PANI-X and G/Ti-PANI-X samples were heat-treated at 900 °C with the aim of obtaining ordered-mesoporous N-doped carbon materials. All of them have also been electrochemically characterized by cyclic voltammetry in nitrogen-saturated 0.5 M H_2_SO_4_ and 0.1 M KOH solutions, using a scan rate of 50 mV·s^−1^. Figure 2 shows the voltammograms of all resultant carbon-based material samples. It also contains the CV for the G/Si and G/Ti samples after heat treatment at 900 °C for comparison purposes. 

The electrochemical response of polyaniline in 0.1 M H_2_SO_4_ solution is lost after the carbonization at 900 °C. Polyaniline decomposes forming a carbon material in which amine and imine species from polyaniline lead to the formation of heterocyclic nitrogen functionalities, such as pyridine, pyrrole, quaternary-type nitrogen, and pyridone functional groups with a very different electroactivity [26]. Figure 2a shows an important decrease in current density of G/Si-PANI-5min-900, in comparison with G/Si-PANI-5min (Figure 1b). In any case, the resultant carbon-based composites exhibit an enhancement in the electrochemical response compared to the G/Si sample (Figure 2a). This should be related to the presence of a thin film formed by the N-doped carbon material on the inorganic matrix.

However, in the case of the Titania template, the electrochemical response is lower in G/Ti_PANI_5min_900 than for the pristine G/Ti (Figure 2b). Taking into account that the synthesized Titania thin film is mainly formed by anatase phase [30] and that the transformation of anatase into rutile takes place at around 500 °C [37], the most likely explanation of the voltammetric behavior is the loss of surface area through the anatase–rutile transformation during the heat treatment at 900 °C, since rutile is known to have a much lower BET surface area than anatase [38]. This aspect is corroborated in the next section. Furthermore, the amount of electropolymerized polyaniline does not seem to have an effect on the CV profile of the heat-treated samples, which might be because of the blocking of the resultant N-doped carbon materials within the rutile-phase Titania structure.

The electrochemical characterization of G/Si-PANI-X_900 and G/Ti-PANI-X_900 samples was also performed in a nitrogen-saturated alkaline solution. Figure 2c,d show the cyclic voltammetry of all composites under 0.1 M KOH solution at a scan rate of 50 mV·s^−1^. Once again, the higher the amount of polyaniline in G/Si, the higher the amount of the resultant carbon material. Interestingly, a new redox process is observed in the profile of G/Si-PANI-5min-900 that does not appear in the other samples, and that it is neither a characteristic peak for silica nor for carbon materials. This peak might be attributed to a new species derived from a reaction between polyaniline and the silica film during the carbonization treatment. Indeed, the presence of M–O–N oxynitride species has been reported as responsible for a redox peak in alkaline solution at a potential similar to the observed in this sample [39]. A redox process is also appreciable in G/Ti-PANI-X-900 samples, although it is smaller and is shifted to less positive potentials than the observed in G/Si-PANI-5 min-900 sample. The appearance of the mentioned redox processes is due to the reaction between the polyaniline and the inorganic film during the heat treatment. This was confirmed, since G/Ti and G/Si heat-treated at 900 °C in the absence of PANI do not present such redox processes in alkaline media (Figure 2e).

### 3.3. Physicochemical Characterization

FESEM images are included in Appendix A. FESEM images show a smoother surface of the G/Si and G/Ti samples (Appendix A, respectively) compared with the macroporous G (Appendix A), which confirm the deposition of Silica and Titania thin films. However, this technique does not provide information about the porous structure. TEM images are presented in Figure 3. Figure 3a,b show the ordered mesoporous structures of the Silica and Titania thin films with a pore size close to 10 nm in both cases. No significant changes were observed once the chronoamperometric polymerization was done (Figure 3c,d for G/Si_PANI_5min and G/Ti_PANI_5min, respectively), which suggests the presence of a thin film of polyaniline within the porous structure of Silica and Titania. Since the porous structure is observable, the total blocking of the porosity can be discarded. This confirms the creation of a 2D-ordered thin film of polyaniline within the porosity of both templates.

Once the heat treatment is applied, the pristine ordered mesoporosity of the Silica film is still observed (Figure 3e); however, the Titania template completely loses the presence of mesopores and forms small nanoparticles as a consequence of the change in the Titania structure (Figure 3f). Indeed, a higher resolution of TEM images confirms the presence of nanoparticles with a hexagonal morphology of around 25 nm of size (Figure 3g). 

The formation of these small nanoparticles is in agreement with the anatase–rutile phase transformation with the heat treatment that results in a loss in surface area of the Titania template. FESEM images (Appendix A) also confirm the formation of these nanoparticles. XRD analysis has also been carried out for the samples obtained from Titania. Unfortunately, since the amount of Titania in the final composite is too small, XRD profiles (Appendix A) do not show the characteristics peaks of the titanium oxide, and only a graphite profile is observed. However, XRD of bulk titania, where this transformation is clearly observed after the heat treatment, can be found in the literature [37]. 

All samples were also studied by XPS analysis. Figure 4 shows the N1s spectra for all materials where PANI has been electropolymerized. C1s and O1s spectra of all materials are included in Appendix A. G/Si-PANI-X and G/Ti-PANI-X samples show two peaks in the N1s spectra, which are associated with polyaniline [40,41,42]: (i) neutral amine groups at 399.5 eV and (ii) positively charged nitrogen atoms at approximately 401.7 eV. Table 2 shows the ratio between positively charged nitrogen atoms and total nitrogen content and the total nitrogen atomic content. In G/Si-PANI-X samples, the higher the polymerization time, the higher the contribution of positively charged N atoms. The trend is similar for G/Ti-PANI-X samples. 

Concerning the heat-treated samples, Figure 5 shows the N1s spectra of all G/Si-PANI_X_900 and G/Ti-PANI_X_900 samples. All materials show two peaks, which are related to pyrrolic/pyridonic nitrogen species, at approximately 400.1 eV [43,44] and the presence of quaternary-type nitrogen species at 401.2 eV [43,44]. However, in G/Ti_PANI_X_900 samples, there is a third peak at 402 eV, which can be associated to oxidized nitrogen (N–O species) [45,46].

Generally, all samples are highly enriched in quaternary-type nitrogen functional groups, but specifically, in G/Si_PANI_X_900 samples, it is observed that the higher the polymerization time, the higher the amount of quaternary-type nitrogen species after the heat treatment. In fact, G/Si_PANI_5min_900 sample mainly contains quaternary-type nitrogen species. 

Table 2 shows the nitrogen content of all materials obtained from XPS. In both templates, the higher the amount of polyaniline (Table 1), the higher the nitrogen content of the resultant N-doped carbon composite. After the heat treatment, there is an important decrease in the nitrogen content as a consequence of the carbonization of the polymeric chains. The yield of the carbonization can be determined from the nitrogen content before and after the heat treatment, being around 35%.

Moreover, it should be highlighted that the shortest period of polymerization time (10 s), i.e., lower amount of polymer, leads to an amount of nitrogen after the high-temperature treatment, which is within the sensitivity of the technique (less than 0.2 at.%). This seems to indicate that the dimers or oligomers of low molecular weight are volatilized during the heat treatment and do not produce significant amount of carbon material.

### 3.4. Electrocatalytic Activity

The electrocatalytic activity of all materials was studied in O_2_-saturated 0.1 M KOH solution. The steady-state cyclic voltammograms in O_2_-saturated 0.1 M KOH solution of G/Si_PANI-5min-900 and G/Ti_PANI-5min-900 are shown in Figure 6a,b, respectively. The large reduction current density and the tilted profile, in comparison with the N_2_-saturated solution profile, are characteristic features of catalytic materials towards oxygen reduction reaction. All materials (not included in Figure 6 for the sake of clarity) showed similar behavior. To get further insights, Figure 6c,d show the LSV curves for all samples towards the oxygen reduction reaction in which the current of the electrochemical double layer has been subtracted (using the LSV in absence of O_2_). G/Si_900 and G/Ti_900 samples were included for comparison purposes. Moreover, we have included LSV curves of a commercial Pt/C deposited on the macroporous graphite (loading: 0.4 mg·cm^−2^).

Bare graphite shows the poorest catalytic activity towards oxygen reduction reaction with an onset potential close to 0.80 V vs. RHE. The addition of a thin film of Silica seems to enhance the catalysis of this reaction up to 0.83 V (onset potential). Then, the heat treatment of the G/Si composite (without any polyaniline) improves the onset potential of the materials up to 0.87 V. The highly effective catalysis of silica-doped carbon has already been demonstrated from experiments and theory [47]. Si can change the charge distribution of the carbon framework and also the adsorption mode of the oxygen molecule [48]. 

Once the polyaniline is electropolymerized and heat-treated within the porosity of SiO_2_, relevant changes are observed. G/Si-PANI-10s-900 does not exhibit differences with respect to G/Si-900, probably because the amount of N-doped carbon material is very low. However, higher polymerization time, and then higher amount of polyaniline, leads to the formation of well-formed polyaniline, as electrochemical measurements and XPS analysis show. Therefore, the heat treatment leads to the formation of carbon-based composites with a nitrogen content close to 0.5 at.% (see Table 2). Interestingly, G/Si-PANI-30s-900 and G/Si-PANI-5min-900 samples show higher catalytic activity than G/Si- 900, achieving a *E*_ONSET_ close to 0.92 V. 

A similar trend is also observed in G/Ti-PANI_X_900 samples. G/Ti is a highly efficient catalytic sample by itself [49] and the heat treatment does not produce a significant improvement of its electrochemical activity for ORR. Once again, the presence of a small amount of oligomers does not seem to have a positive effect in the catalytic activity of G/Ti_PANI_10s_900, but a slight enhancement in catalytic activity is observed when the well-formed polyaniline is heat-treated at 900 °C. In order to study the stability of the best catalyst, chronoamperometric experiments were performed maintaining the working electrode potential of 0.65 V vs. RHE for 20,000 s. Pt/C catalysts was included for comparison purposes. Appendix A shows the chronoamperometric curves, and it can be observed that G/Si_PANI_5min-900 shows a decrease in the catalytic activity up to 84% after 500 s of activity, but then this current keeps constant in a range of 86–92%. On the other hand, the commercial Pt-based catalysts has a decrease in activity up to 80% after only 2200 s and 71% after 20,000 s, pointing out the high performance of the electrocatalyst prepared in this work. 

Tafel slopes, calculated from Tafel plots (Appendix A), for the different materials provide also interesting information about the ORR mechanism (Table 3). While N-doped carbon materials/Silica composites show slopes close to 80 mV·dec^−1^, the N-doped carbon materials/Titania composites have higher Tafel slopes (120 mV·dec^−1^). The Tafel slope of 120 mV·dec^−1^ is characteristic of reactions whose rate-determining step is the formation of superoxide species involving one electron transfer [50,51], which would be associated with a two-electron pathway and hydrogen peroxide formation. On the other hand, lower Tafel slopes indicate that other chemical reactions participate in the reaction mechanism [50,51,52], which might correspond to an increase in the number of transferred electrons. Therefore, the ORR mechanism in G/Si-PANI-X-900 and G/Ti-PANI-X-900 go through different pathways.

According to the physicochemical and electrochemical characterization, the enhancement in the catalytic activity of the most active samples can be associated with the presence of an N-doped carbon framework within the porous structure of the Silica and Titania, which exhibits a quaternary nitrogen-enriched surface. Indeed, edge-type quaternary nitrogen species have been proposed as highly efficient active sites towards ORR with high selectivity towards water formation due to its low kinetic barriers and bridging binding mode for oxygen molecule chemisorption [21,22,23,24,53]. Nevertheless, the higher amount of N-doped carbon material in the samples obtained with the highest polymerization times can also explain the enhanced electrocatalysis. Furthermore, the 2D-ordered mesoporosity of these composites enables the accessibility of the oxygen molecule and the intermediates of the ORR to efficiently reach the active sites of the N-doped carbon materials. 

An etching process was carried out in order to evaluate the effect of the inorganic template in the electrocatalytic activity of the resultant samples. For that purpose, the most catalytic sample, G/Si_PANI_5min_900, was immersed in a 10 M NaOH solution during 10 h at 60 °C with the aim of removing the silica template. After that, it was washed with water several times. The resultant sample is referred as G/Si-PANI-5min-900_NaOH. Figure 7a shows the comparison between the catalytic activity of the samples before and after the etching. Interestingly, removal of the silica template of the composites leads to a significant decrease in the catalytic activity of the material. This fact should be associated with the modification of the most catalytic nitrogen species. In order to discern whether there is a modification of the nitrogen functional groups due to the action of the high concentrated sodium hydroxide solution, XPS analysis after etching was performed. Figure 7b shows the N1s spectra of G/Si_PANI-5min-900_NaOH, where important changes are observed. The quaternary-type nitrogen species of G/Si_PANI_5min_900 sample have transformed into, mainly, pyridone species after the etching process. This transformation can only be understood if the quaternary-type nitrogen species are located at the edge of the carbon framework, since the oxidation of a basal-plane quaternary nitrogen species is thermodynamically unfavorable [53]. The results are in agreement with the higher catalytic activity of edge-type quaternary nitrogen species with respect to pyridone-type nitrogen functional groups. Even though the etching process has diminished the catalytic activity of the sample, the performance is still highly efficient, since the pyridonic-like groups have also been proposed as active sites in N-doped carbon materials [15,20].

## 4. Conclusions

Two-dimensional-ordered polyaniline thin film was successfully prepared through the potentiostatic polymerization of aniline within the porous structure of two ordered mesoporous templates: Silica and Titania. The polymerization time permits to control the amount of polymer electrodeposited on the surface of the templates. Moreover, XPS reveals an increase in the positively charged nitrogen species with the polymerization time and a maximum nitrogen content of 1.2 at.%, obtained in G/Ti_PANI_5min composite.

The excellent control over the polyaniline amount leads to a good control over the resultant carbon materials composites after heat-treated samples when silica is used as a template. However, the Titania phase transformation from anatase to rutile hampers the benefits of the obtained N-doped carbon materials/Titania composites. In fact, TEM images confirm the loss of surface area because of a morphological transformation of the anatase to rutile phase. On the other hand, well-ordered mesoporosity in the carbon material composites was maintained using Silica as a template. It is observed from N1s spectra that the nitrogen species are mainly quaternary-type nitrogen functional groups. 

The catalytic activity of the heat-treated samples towards oxygen reduction reaction was carried out in an alkaline solution. Although all samples showed an enhancement after the carbonization stage, those materials where Silica acts as template exhibited highly effective electrocatalysis, with an *E*_ONSET_ potential higher than 0.9 V vs. RHE and lower Tafel plot. Once the Silica template was removed with highly concentrated alkaline solution, the electrocatalytic activity of the samples decreases as a consequence of the transformation of edge-type quaternary nitrogen species into pyridonic nitrogen functional groups, pointing out the crucial role that quaternary nitrogen species play in ORR electrocatalysis.

## Figures and Tables

**Figure 1 polymers-12-02382-f001:**
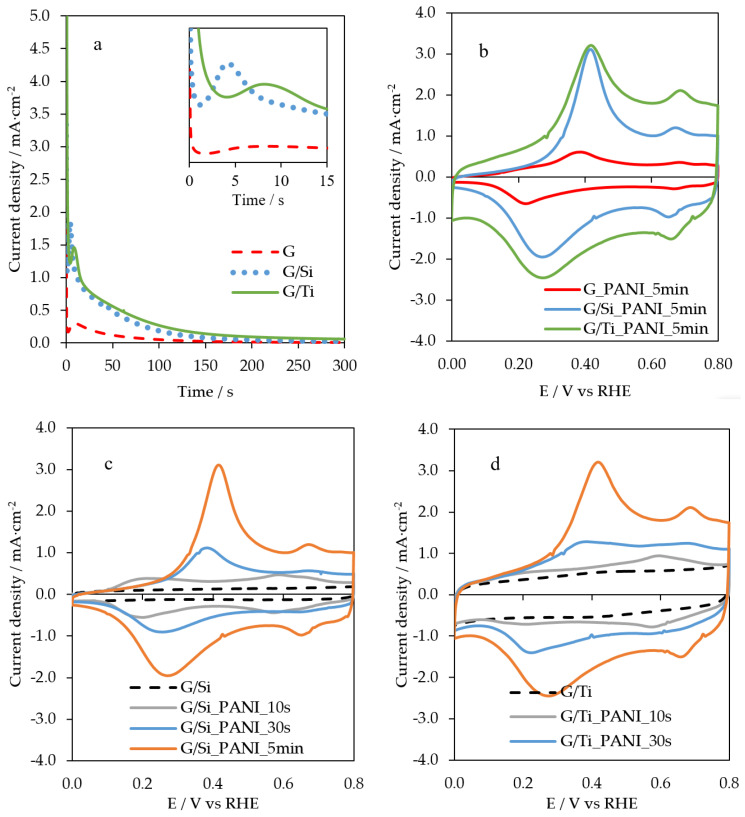
(**a**) Chronoamperometric profiles of aniline polymerization at 0.95 V vs. RHE during 5 min on G, G/Si, and G/Ti, (**b**) CV profiles of the resultant samples (G, G/Si, G/Ti) after 5 min of aniline polymerization, (**c**,**d**) CV profiles of all G/Si-PANI-X and G/Ti-PANI-X samples, respectively. Scan rate = 50 mV·s^−1^; 0.1 M H_2_SO_4_.

**Figure 2 polymers-12-02382-f002:**
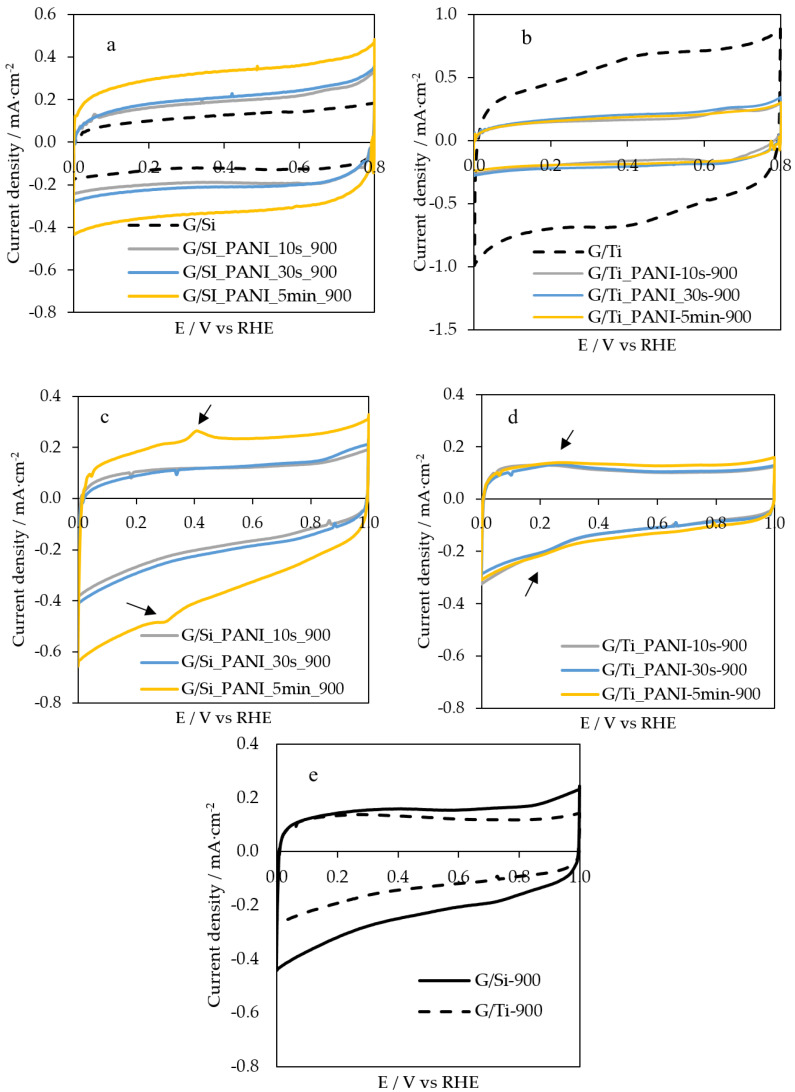
CV profiles of (**a**) G/Si-PANI-5min and (**b**) G/Ti-PANI-5min after the heat treatment in 0.1 M H_2_SO_4_ solution. CV profiles of the heat-treated (**c**) G/Si-PANI_X_900, (**d**) G/Ti-PANI-X-900, and (**e**) G/Si-900 and G/Ti-900 samples in 0.1 M KOH solution. Scan rate = 50 mV·s^−1^.

**Figure 3 polymers-12-02382-f003:**
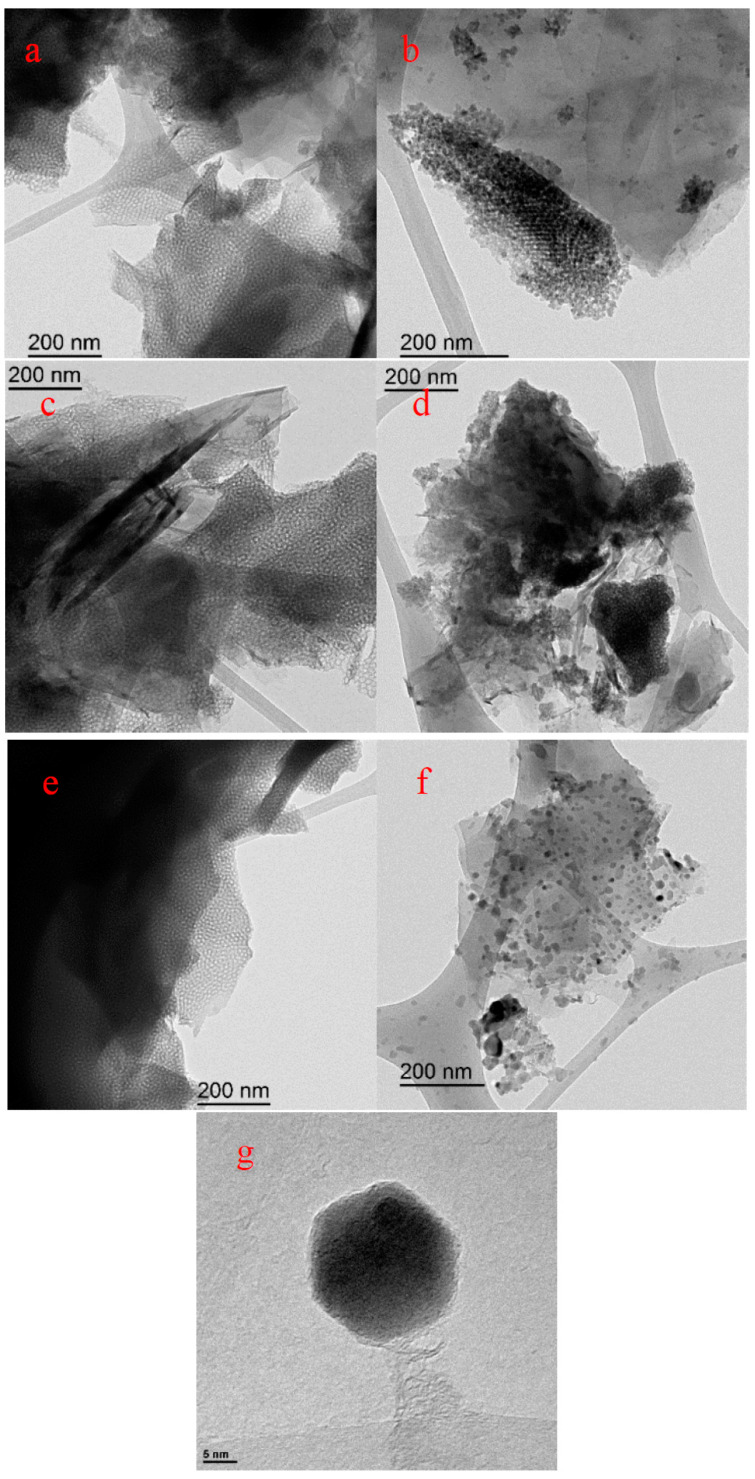
TEM images of (**a**) G/Si, (**b**) G/Ti, (**c**) G/Si_PANI_5min, (**d**) G/Ti_PANI_5min, (**e**) G/Si_PANI_5min_900, (**f**) G/Ti_PANI_5min_900, and (**g**) high resolution of G/Ti_PANI_5min_900.

**Figure 4 polymers-12-02382-f004:**
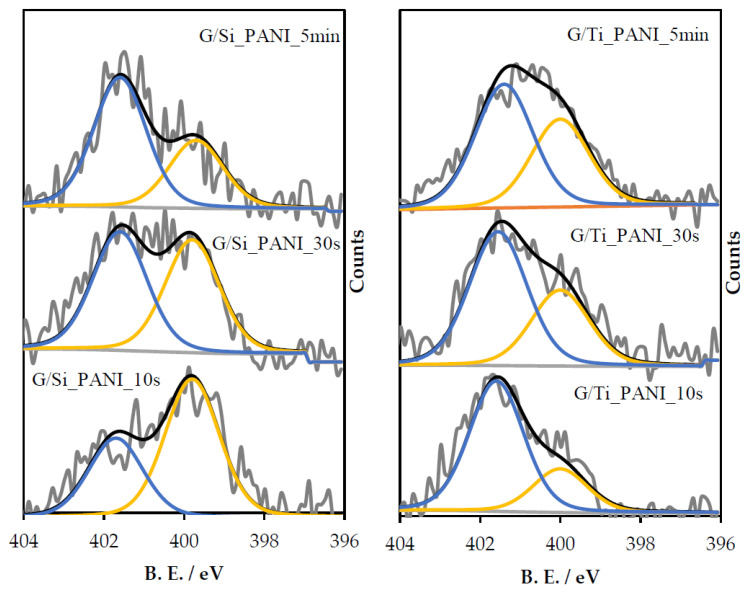
N1s spectra of all resultant G/Si-PANI-X and G/Ti-PANI-X samples. The blue line represents positively charged nitrogen, whereas the yellow line is associated with neutral amine species.

**Figure 5 polymers-12-02382-f005:**
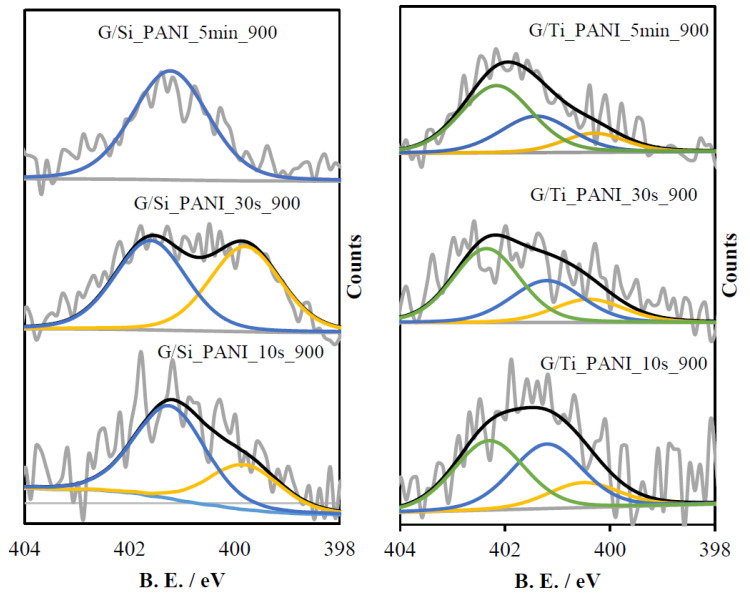
N1s spectra of all heat-treated G/Si-PANI-X-900 and G/Ti-PANI-X-900 samples.

**Figure 6 polymers-12-02382-f006:**
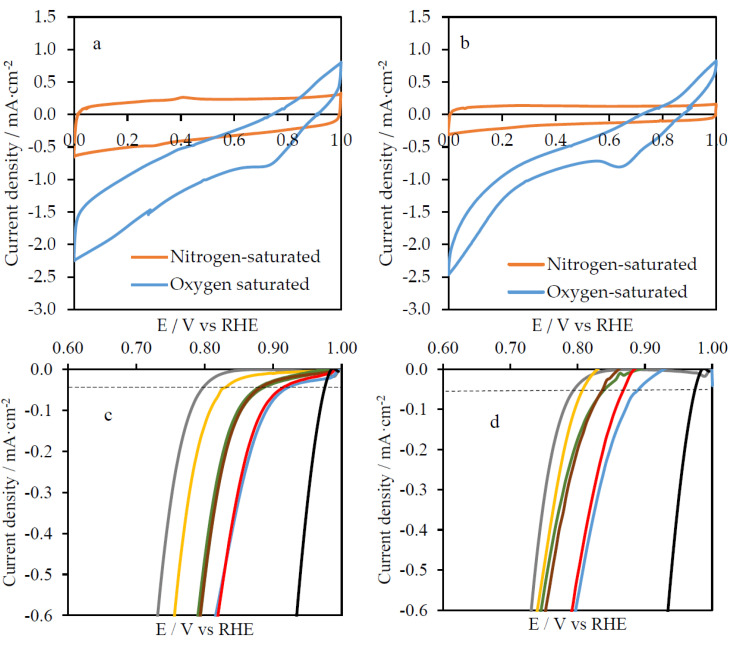
Steady voltammograms in nitrogen- and oxygen-saturated 0.1 M KOH solution at 50 mV·s^−1^ of (**a**) G/Si-PANI-5min-900 and (**b**) G/Ti-PANI-5min-900. Linear sweep voltammetry curves in O_2_-saturated 0.1 M KOH solution at 50 mV·s^−1^ for (**c**) G/Si-PANI-X-900 and (**d**) G/Ti-PANI-X-900 samples. In linear sweep voltammetry (LSV) curves, grey represents G, yellow represents G/X, green represents G/X_900, brown represents G/X_PANI_10s_900, blue represents G/X-PANI_30s_900, and red represents G/X_PANI_5min_900, with X being (**c**) Si or (**d**) Ti. Black line represents the commercial Pt/C catalysts, included for comparison purposes.

**Figure 7 polymers-12-02382-f007:**
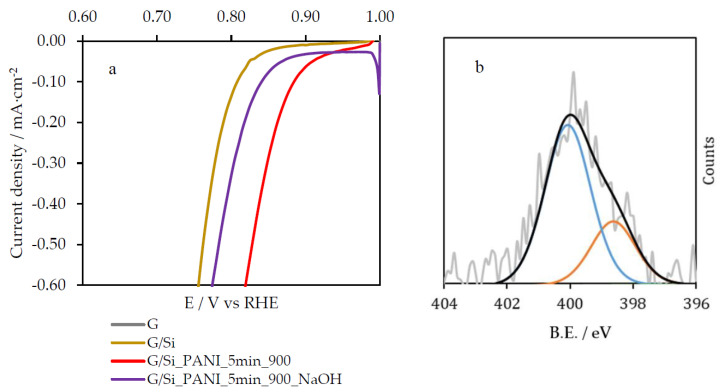
(**a**) Linear sweep voltammetry curves of G/Si_PANI_5min_900_NaOH in O_2_-saturated 0.1 M KOH solution at 50 mV·s^−1^ and (**b**) N1s from XPS analysis of G/Si_PANI_5min_900_NaOH; blue peak represents pyridonic-like groups and orange line represents pyridine species.

**Table 1 polymers-12-02382-t001:** Charge obtained from the potentiostatic pulse and the calculated amount of electrodeposited polyaniline on each electrode.

Electrode	Time	Q/mC	Polyaniline Amount */µg
G_PANI	10 s	2.64	2
	30 s	7.43	7
	5 min	18.25	17
G/Si_PANI	10 s	13.80	13
	30 s	30.39	29
	5 min	70.92	67
G/Ti_PANI	10 s	19.83	19
	30 s	38.31	36
	5 min	93.95	88

***** Assuming a value of 1 electron per deposited aniline monomer.

**Table 2 polymers-12-02382-t002:** Ratio between positively charged nitrogen species and total nitrogen content and the atomic percentage of the nitrogen content obtained from XPS.

Sample	N^+^/N_total_ Ratio	N Content/at.%
G/Si_PANI_10s	0.39	0.3
G/Si_PANI_30 s	0.49	0.6
G/Si_PANI_5 min	0.64	0.7
G/Ti_PANI_10s	0.25	0.6
G/Ti_PANI_30 s	0.36	0.8
G/Ti_PANI_5 min	0.42	1.2
G/Si_PANI_10 s_900	-	<0.2
G/SI_PANI_30 s_900	-	0.2
G/Si_PANI_5 min_900	-	0.4
G/Ti_PANI_10 s_900	-	<0.2
G/Ti_PANI_30 s_900	-	0.3
G/Ti_PANI_5 min_900	-	0.5

**Table 3 polymers-12-02382-t003:** Electrochemical parameters calculated from LSV curves of the different electrocatalysts in O_2_-saturated 0.1 M KOH solution at 50 mV·s^−1^.

Sample	*E*_ONSET_/V vs. RHE (−0.05 mA·cm^−2^)	Tafel Slope/mV·dec^−1^
G	0.80	73
G/Si	0.83	65
G/Si_900	0.87	77
G/Si_PANI_10 s_900	0.87	80
G/Si_PANI_30 s_900	0.92	89
G/Si_PANI_5 min_900	0.91	80
G/Ti	0.81	84
G/Ti_900	0.84	110
G/Ti_PANI_10 s_900	0.84	131
G/Ti_PANI_30 s_900	0.89	118
G/Ti_PANI_5 min_900	0.87	124

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
