# Peer review of "Polyaniline-Derived N-Doped Ordered Mesoporous Carbon Thin Films: Efficient Catalysts towards Oxygen Reduction Reaction"

_polymers, 2020, doi:10.3390/polym12102382_

Round 1

Reviewer 1 Report

This paper demonstrated a new electrocatalyst based on N-doped ordered mesoporous carbon thin film and its electrocatalytic ability towards ORR. The N-doped mesoporous carbon was synthesized from a graphene and polyaniline composite using heat treatment. The film was analyzed by electrochemical methods, XPS, and TEM. The G/Si(or Ti)_PANI_5min_30s (or 5min)_900 shows good electocatalytic activity for ORR. The experiment was conducted logically, and the results of the experiment were explained well. Therefore, it seems ready for publication, after a few reinforce.

Comments and questions to reinforce the manuscript:

  1. The abbreviation "G" should be mentioned in line 74.
  2. The electrochemically synthesized PANI is decomposed to various anime derivatives such as pyridine, pyrrole, etc. after 900oC of heat treatment. Why don’t you directly add these amine derivatives including aniline monomer to the graphene film without the pain electrochemical polymerization step? Explain the need for polymer formation step.
  3. In Figure 6, the blue line in 6a or 6b are exactly same experimental conditions with red lines in 6c and 6d. However, the slope of the curves did not look the same. Did the RDE was used in 6c and 6d? The curve looks like that of RDE not that of CV.

Author Response

This paper demonstrated a new electrocatalyst based on N-doped ordered mesoporous carbon thin film and its electrocatalytic ability towards ORR. The N-doped mesoporous carbon was synthesized from a graphene and polyaniline composite using heat treatment. The film was analyzed by electrochemical methods, XPS, and TEM. The G/Si(or Ti)_PANI_5min_30s (or 5min)_900 shows good electocatalytic activity for ORR. The experiment was conducted logically, and the results of the experiment were explained well. Therefore, it seems ready for publication, after a few reinforce.

Comments and questions to reinforce the manuscript:

1. The abbreviation "G" should be mentioned in line 74.

We thank the reviewer by his/her comment. The abbreviation has been indicated in the new version of the manuscript. See line 75 in the revised version.

MANUSCRIPT AMENDED

2. The electrochemically synthesized PANI is decomposed to various anime derivatives such as pyridine, pyrrole, etc. after 900oC of heat treatment. Why don’t you directly add these amine derivatives including aniline monomer to the graphene film without the pain electrochemical polymerization step? Explain the need for polymer formation step.

Polyaniline is an excellent precursor of carbon materials with high N content after heat treatment with a good yield. The large polymer chains lead to the formation of N-doped carbon materials through condensation reactions. This means that in one step we can control the N content as well as the chemical nature of the N species present in the carbon thin film deposited on the silica template. On the other hand, the addition of aniline without any other treatment would not produce a significant incorporation of N species strongly bonded to the graphene film. Thus, aniline monomers would dissolve in the solution or would volatilize if any heat treatment is applied.

We have added a comment in the revised version of the manuscript (see line 62 in the revised version)

MANUSCRIPT AMENDED

3. In Figure 6, the blue line in 6a or 6b are exactly same experimental conditions with red lines in 6c and 6d. However, the slope of the curves did not look the same. Did the RDE was used in 6c and 6d? The curve looks like that of RDE not that of CV.

We would like to thank the reviewer by his/her comment, which has helped us to clarify this statement in Figure 6 in the new version of the manuscript. Figure 6a and b shows the CV of G/Si-PANI-5min-900 and G/Ti-PANI-5min-900. However, Figure 6c and d shows the LSV curves related to the ORR, in which the current associated with the electric double layer (obtained from LSV in the absence of O2) is subtracted from the total current, as follows;

jTotal = jEDL + jORR

jORR = jTotal - jEDL

Nevertheless, this has been explained in the new version of the manuscript (see line 317 in the revised version).

MANUSCRIPT AMENDED.

Reviewer 2 Report

The authors report the preparation of polyaniline-derived N-doped carbon thin films on silica and titania template as an efficient catalyst for ORR in alkaline electrolyte. I think there is a lack of concrete evidence on some results and conclusions. Additional characterization and discussion are needed in this manuscript for more clarification and confirmation. The authors should address the comments and suggestions listed below:

  1. The font type and style used in all graphs in the manuscript should be identical.
  2. The SEM observation on all samples is needed to confirm the morphology and porous structure of silica and titania templates as well as X-PANI-900 on the graphite sheet.
  3. The authors claimed that the loss of surface area and mesoporosity of the titania template was due to anatase-rutile phase transformation at high-temperature treatment (900 oC). I suggest that the authors perform XRD measurements on the titania before and after heat-treatment to confirm the presence of anatase and rutile phases as evidence.
  4. The peak position of quaternary type and positively-charged nitrogen atoms are located at nearly the same binding energy. How to distinguish these two nitrogen species from XPS deconvolution?
  5. For the electrochemical measurements, the current density should also be considered into consideration for ORR activity in addition to onset potential.
  6. The LSV curve of a commercial Pt/C should be added in Figures 6c and 6d for comparison.
  7. The authors should provide more discussion on the ORR mechanism on the catalysts that proceed via a two or four-electron pathway. The RDE measurement concomitant with the K-L plot analysis is needed for this discussion.
  8. Please show the Tafel plot of all samples in the manuscript or Supporting Information.
  9. In addition to the analysis of XPS N 1s, XPS C 1s, and O1s spectra should also be revealed and described. The authors may add these results to Supporting Information.
  10. The different ORR activity of G/X-PANI at various polymerization times may be due to the difference in the amount of N-doped carbon coated on silica and titania template, implying a difference in catalyst loading for each polymerization time. It is known that catalyst loading on the electrode has a strong influence on the ORR catalytic activity in terms of both onset potential and current density.
  11. The accelerated degradation test or I-t chronoamperometry should be carried out to confirm its durability under long-term operation.

Author Response

The authors report the preparation of polyaniline-derived N-doped carbon thin films on silica and titania template as an efficient catalyst for ORR in alkaline electrolyte. I think there is a lack of concrete evidence on some results and conclusions. Additional characterization and discussion are needed in this manuscript for more clarification and confirmation. The authors should address the comments and suggestions listed below:

1. The font type and style used in all graphs in the manuscript should be identical.

We are sorry about the mistakes. We hope that the new version of the manuscript includes the correct font type and style.

2. The SEM observation on all samples is needed to confirm the morphology and porous structure of silica and titania templates as well as X-PANI-900 on the graphite sheet.

FESEM images have been included in the new version of the manuscript as supporting information. Although FESEM images do not provide information about the porosity of the materials, they support the deposition of the templates (see line 246 in the revised version).

MANUSCRIPT AMENDED.

3. The authors claimed that the loss of surface area and mesoporosity of the titania template was due to anatase-rutile phase transformation at high-temperature treatment (900 oC). I suggest that the authors perform XRD measurements on the titania before and after heat-treatment to confirm the presence of anatase and rutile phases as evidence.

We would like to thank the reviewer by his/her comment. We have done XRD measurements. Unfortunately, the amount of Titania present in the composite is too small and the characteristic peaks of the titanium oxide are not observed in the XRD profiles of the samples. Only peaks corresponding to graphite are identified. However, we have included a comment and a reference of a work which shows this phase transition in the XRD of bulk titanium dioxide after heat treatment at 900 ºC in line 264.

MANUSCRIPT AMENDED.

4. The peak position of quaternary type and positively-charged nitrogen atoms are located at nearly the same binding energy. How to distinguish these two nitrogen species from XPS deconvolution?

Quaternary-type nitrogen species and positively charged nitrogen atoms appear at the same binding energy. It is known that polyaniline contains positively charged nitrogen species, imines and amines, being all these functional groups unstable at the temperatures that have been employed during the heat treatment (900ºC). The carbonization of polyaniline has been widely studied in the literature (S. Kuroki et al., Carbon 55 (2013) 160, Z. Rozlívková et al., Synth. Met. 161 (2011) 1122), in which the pristine nitrogen species of the polymer leads to quaternary, N-C-O, pyridine and N-O functional groups, among others. So, even though it is impossible to distinguish between both groups, it is known that positively charged nitrogen species present in polyaniline are unstable in carbon materials, as well as quaternary nitrogen functional groups are not found in the large chains of polyaniline.

5. For the electrochemical measurements, the current density should also be considered into consideration for ORR activity in addition to onset potential.

Limiting current density is a parameter used in the measurement of the catalytic activity towards the oxygen reduction reaction. However, these samples were prepared as sheets, and not a powder, which makes impossible the testing of these materials in RDE or RRDE. Therefore, limiting current densities cannot be obtained.

6. The LSV curve of a commercial Pt/C should be added in Figures 6c and 6d for comparison.

We agree with the reviewer. A thin film of Pt/C was deposited on the same macroporous graphite (G) to compare with the samples obtained in this contribution. The results are included and commented in the new version of the manuscript (see Figure 6, supporting information and discussion).

MANUSCRIPT AMENDED

7. The authors should provide more discussion on the ORR mechanism on the catalysts that proceed via a two or four-electron pathway. The RDE measurement concomitant with the K-L plot analysis is needed for this discussion.

We would like to thank the reviewer by his/her comment. Unfortunately, the samples are prepared as sheets (1x1 cm), which makes impossible their evaluation in RDE or RRDE measurements. Nevertheless, the mechanisms and selectivity have been discussed from Tafel slopes.

8. Please show the Tafel plot of all samples in the manuscript or Supporting Information.

Tafel plots of all samples have been included in the Supporting Information.

MANUSCRIPT AMENDED

9. In addition to the analysis of XPS N 1s, XPS C 1s, and O1s spectra should also be revealed and described. The authors may add these results to Supporting Information.

XPS of C1s and O1s are included in the Supporting Information

MANUSCRIPT AMENDED

10. The different ORR activity of G/X-PANI at various polymerization times may be due to the difference in the amount of N-doped carbon coated on silica and titania template, implying a difference in catalyst loading for each polymerization time. It is known that catalyst loading on the electrode has a strong influence on the ORR catalytic activity in terms of both onset potential and current density.

We would like to thank the reviewer by his/her comment. The amount of N-doped carbon on Silica and Titania can also be responsible for the enhanced catalysts. We have included an explanation about this possibility (see line 374 in the revised version of the manuscript).

MANUSCRIPT AMENDED

11. The accelerated degradation test or I-t chronoamperometry should be carried out to confirm its durability under long-term operation.

We agree with the reviewer. Stability tests have been carried out for the highest efficient sample and the commercial platinum-based electrode was included for comparison purposes (see supporting information). A brief discussion about the experiment has been included in the new version of the manuscript (see line 349).

MANUSCRIPT AMENDED

Round 2

Reviewer 2 Report

The authors have provided adequate responses to all of the questions and comments, excepting the stability test. The chronoamperometric measurement was performed for only 1 h (3600 s), which is too short for a stability test. Therefore, the measurement time of chronoamperometry should be increased, typically ranging from 20000 to 50000 s in literature. I suggest this manuscript be accepted after this minor revision.

Author Response

The authors have provided adequate responses to all of the questions and comments, excepting the stability test. The chronoamperometric measurement was performed for only 1 h (3600 s), which is too short for a stability test. Therefore, the measurement time of chronoamperometry should be increased, typically ranging from 20000 to 50000 s in literature. I suggest this manuscript be accepted after this minor revision.

We thank the reviewer for the comments. Longer chronoamperometric time (20,000 s) was performed for the stability test in the samples G/Si_PANI_5min-900 and Pt/C. The manuscript has been modified from line 349 and Figure S9 has been changed.

MANUSCRIPT AMMENDED